# SPARSELY CONNECTED VARIATIONAL AUTOENCODER FOR SCRNA-SEQ DATA PROCESSING

## ABSTRACT

scRNA-seq presents opportunities to investigate cellular activities at the single-cell level, but data acquisition and its unique characteristics pose distinct challenges in data processing. Traditional tools typically follow a multi-stage workflow involving a series of statistical adjustments and trade-offs. While being straightforward, this approach lacks overall optimization as a single tool, leading biologists to find inconsistent results with different tools. This work introduces `scVAE`, **s**parsely **c**onnected **v**ariational **a**uto**e**ncoder for **s**ingle-**c**ell data processing, an integrated and multi-task tool for scRNA-seq data processing, leveraging self-supervised deep learning techniques to tackle existing challenges. The `scVAE` model includes two key components: a sparsely connected layer utilizes sparse representation to enable the model to operate on the full gene space without performing gene selection, and a *batchified* module learns batch-specific variations with parameterized sub-modules and calculates correction vectors. By reducing data processing overhead, the `scVAE` model improves the overall efficiency with a more streamlined workflow. Experiments and evaluations on real datasets reveal that `scVAE` produces interpretable and biologically meaningful results.

## 1 INTRODUCTION

The success of scRNA-seq experiments has sparked renewed interest among biologists in refining the data processing workflow to facilitate and accelerate biological discoveries. However, scRNA-seq data processing and analysis remain labour-intensive and time-consuming, requiring iterative and validation processes to obtain clear biological insights. Manual interventions are often necessary to assess and adjust the workflow due to potential discrepancies and disagreements. Traditional data analysis workflows typically involve several distinct stages, each with a specific consideration.

Quality control (QC) steps remove low-quality measurements by eliminating cells and genes based on several QC metrics. Data normalization decouples technical variations from biological variations, producing clean biological signals for further processing. Computational methods such as size factor normalization convert raw counts into a fixed value. While these methods are straightforward, Amezquita et al. (2020) demonstrated that they often require substantial expertise.

Gene selection identifies a subset of more informative genes to balance the trade-off between efficiency gains and information loss. The popular `Seurat` highly variable gene (`HVG`) method by Stuart et al. (2019) selects genes based on normalized variance from local polynomial regression. However, different gene selection methods often yield different gene sets.

Cell embedding encodes gene expression profiles into a lower-dimensional space, facilitating cell type annotation and further biological analysis. It also considers batch effect caused by uncontrolled variables. The `Seurat` method by Stuart et al. (2019) adopts a PCA-UMAP hybrid method. Batch effect is removed using the `CCA` method to identify linear correlations in the PCA space.

In contrast to the traditional multi-step workflow, there are integrated approaches utilizing deep learning techniques. Lopez et al. (2018) implemented `scVI`, a hierarchical model with a VAE backend, which significantly improves scalability and performance compared to traditional tools. While the traditional workflow is straightforward, it involves a series of independent adjustments and requires balancing multiple trade-offs, which complicates the overall process. In the meantime, some deep-learning-based methods, such as `scVI`, still rely on elements of the traditional workflow.

## 2 METHODS

### 2.1 scVAE MODEL OVERVIEW

The scVAE model presented in this work is a self-supervised, multi-task neural network model based on the VAE architecture with a zero-inflated negative binomial distribution backend for scRNA-seq data processing. The model operates on the full gene list input, eliminating the need for gene selection and minimizing data processing overhead. This is achieved through two key components to address the unique characteristics of the scRNA-seq data. The sparsely connected (SC) layer learns a sparse representation of the full gene list input, reducing the number of parameters and improving the efficiency for full gene list input. The learned connections also provide insights into gene informativeness. The batchified module (BM) utilizes parameterized sub-modules to encode batch-specific variations and learns correction vectors to remove the batch effect in the dataset. Furthermore, utilizing the learned batch-specific variations, scVAE makes it possible to transfer the *style* of data across batches.

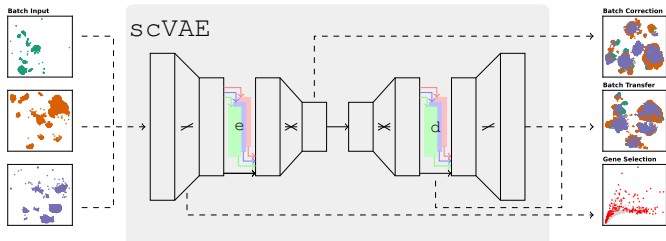

Figure 1: scVAE model overview.

### 2.2 SPARSELY CONNECTED (SC) LAYER

A fully connected (FC) layer, as one of the fundamental neural network layers, connects each input neuron with every output neuron, which requires a substantial number of connections and corresponding parameters, especially when either the number of input or output dimensions is large. For scRNA-seq data, when using the entire gene list as the model input, the first and last FC layers account for approximately 70–80% of the total model parameters. To alleviate this, the SC layer is designed to learn a more efficient sparse representation between the gene space and the hidden space and eliminate the need for explicit gene selection. The SC layer introduces an additional binary matrix $\mathbf{M}_{\mathrm{SC}}$ on a FC layer to control the sparsity. Mathematically, the algebraic operation of a SC layer is:

$$\mathbf{X}_{\mathrm{out}} = \mathrm{SC}(\mathbf{X}_{\mathrm{in}}) = \sigma \left[ \mathbf{X}_{\mathrm{in}} \left( \mathbf{M}_{\mathrm{SC}} \odot \mathbf{W}_{\mathrm{SC}} \right) + \mathbf{b}_{\mathrm{SC}} \right], \tag{1}$$

where $\sigma[\cdot]$ is a nonlinear activation function and $\odot$ represents element-wise multiplication. Learning the parameters of a SC layer involves an iterative process of updating weights and connections. A series of criteria based on quality of reconstruction and magnitude of weights are used to learn the binary mask. Further details on SC layers are outlined in Appendix A.2.

### 2.3 BATCHIFIED MODULE (BM)

The BM module uses multiple copies of parameterized sub-modules to encode batch-specific variations, calculate correction vectors, and address batch effect in multi-batch experiments. In scVAE, BM modules are injected into existing neural network layers, allowing it to learn batch-corrected representations without altering the existing model architecture. The BM module preserves the data dimension, and its output can be interpreted as the batch-corrected version of input data. The number of sub-modules corresponds to the number of batches in the multi-batch experiment.

In both training and testing, input data are split by batch and processed through the corresponding sub-modules, and outputs are concatenated in the same order as the input data. Finally, a ResNet-like skip connection is employed to compute the batch-corrected data. Mathematically, the BM module $g_{\mathrm{bm}}(\cdot)$ can be expressed as:

$$\mathbf{X}_{\mathrm{out}} = \mathrm{BM}(\mathbf{X}_{\mathrm{in}}) = \mathbf{X}_{\mathrm{in}} + \beta_{\mathrm{bm}} \cdot g_{\mathrm{bm}}(\mathbf{X}_{\mathrm{in}}), \tag{2}$$

where $\beta_{\mathrm{bm}}$ is a scalar weighting parameter. The BM module design offers flexibility in the choice of sub-modules. In `scVAE`, sub-modules are implemented with a single FC layer with `TanhShrink` activation function. Further implementation details are outlined in Appendix A.3.

## 3    EXPERIMENTS

To evaluate the performance of `scVAE`, four experiments are designed based on different considerations. For single-batch experiments, cell embedding assesses the model's ability to produce effective lower-dimensional embeddings, while gene selection evaluates the effectiveness and provides biological support for the SC layer. In multi-batch experiments, cell embedding assesses the quality of embeddings with the presence of batch effects, and batch style transfer, an experimental feature, utilizes the BM module to learn and transfer the style of data across batches. The performances are evaluated based on average Silhouette width (ASW), normalized mutual information (NMI), and adjusted Rand index (ARI) with Leiden and $k$-means derived clusters. Results presented in the main text are based on a human triple-negative breast cancer (TNBC) dataset generated by Wu et al. (2020). All visualizations presented in this work are coloured by annotated cell type with details given in Appendix A.1.3. Standard errors are calculated over 20 random model initializations.

### 3.1    SINGLE-BATCH CELL EMBEDDING

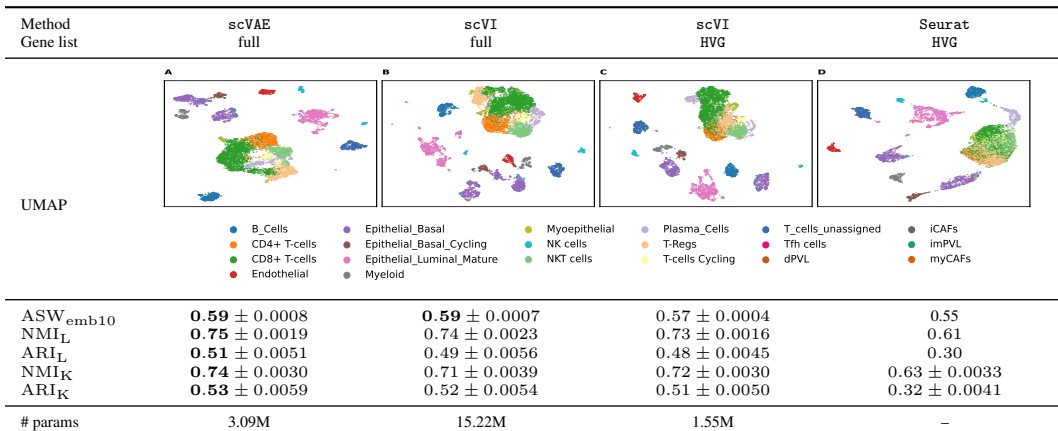

| Method | scVAE | scVI | scVI | Seurat |
| --- | --- | --- | --- | --- |
| Gene list | full | full | HVG | HVG |
| UMAP | | | | |
| $\mathrm{ASW}_{\mathrm{emb10}}$ | **0.59** $\pm$ 0.0008 | **0.59** $\pm$ 0.0007 | 0.57 $\pm$ 0.0004 | 0.55 |
| $\mathrm{NMI}_{\mathrm{L}}$ | **0.75** $\pm$ 0.0019 | 0.74 $\pm$ 0.0023 | 0.73 $\pm$ 0.0016 | 0.61 |
| $\mathrm{ARI}_{\mathrm{L}}$ | **0.51** $\pm$ 0.0051 | 0.49 $\pm$ 0.0056 | 0.48 $\pm$ 0.0045 | 0.30 |
| $\mathrm{NMI}_{\mathrm{K}}$ | **0.74** $\pm$ 0.0030 | 0.71 $\pm$ 0.0039 | 0.72 $\pm$ 0.0030 | 0.63 $\pm$ 0.0033 |
| $\mathrm{ARI}_{\mathrm{K}}$ | **0.53** $\pm$ 0.0059 | 0.52 $\pm$ 0.0054 | 0.51 $\pm$ 0.0050 | 0.32 $\pm$ 0.0041 |
| # params | 3.09M | 15.22M | 1.55M | – |

Table 1: Quantitative and benchmark results of single-batch cell embedding experiment.

Table 1 includes benchmark analysis between `scVAE`, `scVI`, and `Seurat` in processing single-batch data. Results presented in Table 1 indicate that the `scVAE` model outperforms the `Seurat` method by a large margin, achieving both better embedding quality and improved clustering results. The improvement becomes less pronounced compared to the `scVI`-full and `scVI`-HVG methods. Visually, the `scVI`-full method reveals poorer embeddings in certain clusters, and the `scVI`-HVG method shows increased overlap within sub-clusters. Additionally, `Seurat` fails to clearly separate certain sub-types in the embedding space. In the single-batch experiment, the `scVAE` model can identify an effective lower-dimensional embedding, while significantly reducing the number of parameters compared to state-of-the-art methods.

### 3.2    SINGLE-BATCH GENE SELECTION

`scVAE` introduces `scScore`$_{\mathrm{n}}$ to reflect the gene informativeness through averaging the number of connections assigned in the SC layer over $n$ repetitions and normalizing to $[0, 1]$. The `scScore`$_{\mathrm{n}}$ is analogous to the normalized variance measure as in the `HVG` method. Figure 2 displays four peaks: two above average centred around 1.00 and 0.70; one at the initial starting point centred around 0.34; and one below average centred around 0.00. Although `scVAE` does not perform gene selection, the `scScore`$_{\mathrm{n}}$ derived from the trained SC layer serves as a proxy for the gene informativeness and can be utilized for ranking and selection.

### 3.2.1 scScore$_{20}$ ROC-LIKE ANALYSIS

A receiver operating curve (ROC)-like curve is generated by iterating through scScore$_{20}$. As shown in Figure 2, gene selected based on scScore$_{20}$ consistently outperforms the HVG method across the entire spectrum. The Pearson correlation between scScore$_{20}$ and scaled mean is $\rho_\mu = 0.3886$, and between scScore$_{20}$ and scaled variance $\rho_{\text{Var}} = 0.3228$, both indicating a low to moderate level of correlation. This suggests that the scScore$_{20}$ method outperforms the HVG method in identifying more informative genes for defining cell type clusters, while not being strongly biased toward the expression mean or variance.

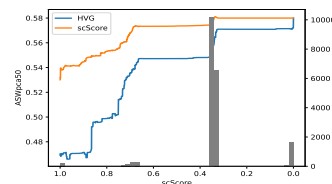

Figure 2: ROC-like analysis.

### 3.2.2 scScore$_{20}$ OVERLAP WITH CANONICAL MARKER GENES

| Annotation Marker | T *CD3D* | CD8 T *CD8A* | T-help *CXCL13* | T-reg *FOXP3* |
|---|---|---|---|---|
| scScore$_{20}$ | 0.69 | 0.71 | 1.00 | 0.69 |
| Annotation Marker | NK *GNLY* | B *MS4A1* | Plasma *JCHAIN* | Myeloid *CD68* |
| scScore$_{20}$ | 0.71 | 0.66 | 1.00 | 0.74 |

scScore *at initialization is 0.34, which corresponds to the highest peak in Figure 2.*

Table 2: scScore$_{20}$ assigned to marker genes.

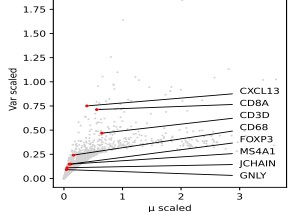

Figure 3: Expression of marker genes.

To further investigate the biological significance of highly-scored genes identified by the scScore$_{20}$ method, the overlap between these genes and known marker genes provided in the corresponding research is studied. Table 2 shows that most of the marker genes have a scScore$_{20}$ score that corresponds to the first or second high-valued peaks in Figure 2, which suggests that the scScore$_{20}$ method considers most underlying canonical marker genes as more informative genes in the self-supervised setting. Figure 3 displays the scaled expression means and variances of canonical marker genes, where some of the marker genes do not necessarily have high expression mean or variance. This experiment reveals that scVAE improves overall performance by identifying biologically meaningful genes. The SC layer does not show strong bias towards empirical statistics, highlighting the advantage over a standalone gene selection method in the workflow and suggesting its potential for identifying unknown marker genes in new datasets.

### 3.2.3 COMBINE scScore$_n$ WITH BENCHMARK METHODS

| Method Gene list | scVI scScore$_{20} \geq 0.60$ | scVI HVG | Seurat scScore$_{20} \geq 0.60$ | Seurat HVG |
|---|---|---|---|---|
| UMAP | | | | |
| ASW$_{\text{emb10}}$ | **0.59** $\pm$ 0.0009 | 0.56 $\pm$ 0.0005 | **0.56** | 0.52 |
| NMI$_{\text{L}}$ | **0.73** $\pm$ 0.0019 | 0.70 $\pm$ 0.0026 | **0.73** | 0.70 |
| ARI$_{\text{L}}$ | **0.50** $\pm$ 0.0052 | 0.41 $\pm$ 0.0065 | **0.48** | 0.42 |
| NMI$_{\text{K}}$ | **0.72** $\pm$ 0.0021 | 0.68 $\pm$ 0.0030 | **0.68** $\pm$ 0.0040 | 0.60 $\pm$ 0.0073 |
| ARI$_{\text{K}}$ | **0.52** $\pm$ 0.0043 | 0.45 $\pm$ 0.0056 | **0.47** $\pm$ 0.0058 | 0.28 $\pm$ 0.0106 |

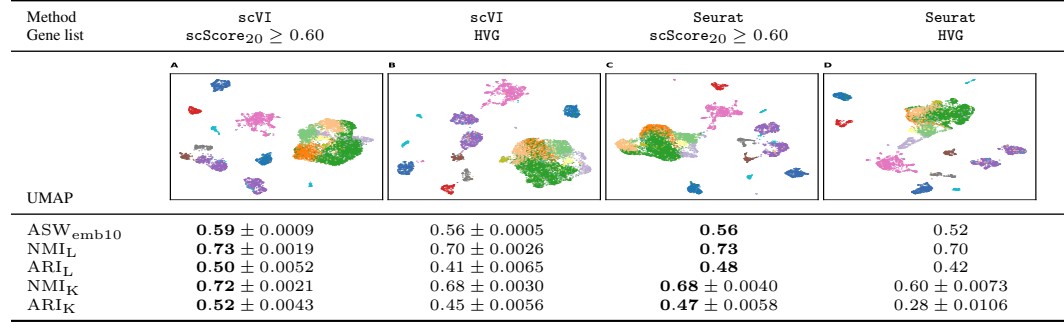

Table 3: Combining scScore$_{20}$ with benchmark methods.

Table 3 shows the quantitative analysis of two benchmark methods using different gene lists as model input. Genes with a scScore$_{20} \geq 0.60$ are selected, resulting in approximately 1,100 genes, which correspond to the two high-valued peaks in Figure 2. Using scScore$_{20}$ for gene selection demonstrates advantages over the HVG method in identifying more informative genes for subsequent analysis. An additional experiment combines scScore$_{20}$ with benchmark methods to investigate its potential benefits to existing approaches. The scVI and Seurat methods are evaluated using highly-scored genes identified by both scScore$_{20}$ and HVG methods as input. Results in Table 3 reveal improved performance across all evaluation criteria when using scScore$_{20}$ identified genes as input compared to HVG. This cascading effect further supports the effectiveness of the SC layer.

## 3.3 MULTI-BATCH CELL EMBEDDING

| Method | scVI | scVI | scVI | Seurat-CCA |
|---|---|---|---|---|
| Gene list | full | full | HVG-batch | HVG-batch |
| | A | B | C | D |
| UMAP | | | | |

Table 4: Benchmark results of multi-batch cell embedding experiment.

The current implementation of SC layer adopts multiple independent masks $\mathbf{M}_{SC}^b$ on the same weight matrix $\mathbf{W}_{SC}$ in a multi-batch setting without information sharing across masks. At this stage, this work only presents preliminary qualitative results in a multi-batch experiment, while continuing to improve on the model design to integrate information from batch-specific masks. Qualitative results above suggest that the scVAE model produces effective batch correction results for most cell types in the dataset. However, the separation between different cell sub-types becomes less pronounced compared to the single-batch experiment due to the increasing diversity arising from different batches. Preliminary quantitative analysis reveals better performance over the traditional Seurat-CCA method and on-par performance with scVI methods.

## 3.4 MULTI-BATCH STYLE TRANSFER

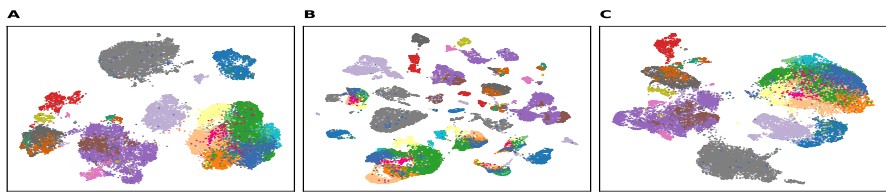

Figure 4: Visualizations of batch style transfer experiment.

This experiment utilizes the trained BM module to transfer batch-specific variations across different batches by activating a single sub-module within the trained decoder BM module and applying correction vectors to all batches. Figure 4 A displays the UMAP visualization of the embedding space shared by both cell embedding and batch transfer experiments. Figure 4 B – C are UMAP visualizations of the full gene space of cell embedding and batch style transfer experiments, respectively. For cell embedding, the decoder BM module re-introduces the batch effect to match the input data such that the batch effect can be observed in Figure 4 B. For the batch style transfer experiment, only a single sub-module is activated such that the final reconstruction follows the style of a specific batch and does not display strong batch effect. Further details are included in Appendix A.4. These observations validate the design assumptions and explain the principle behind BM modules for batch correction. It also offers opportunities for biologists to transfer the style of existing data without re-designing experiments.

## 4 CONCLUSION

This work presents the scVAE model with sparsely connected layers and batchified modules designed for scRNA-seq data processing. The model emphasizes the use of deep learning techniques over traditional statistical adjustments for high-dimensional data handling. Through a series of experiments and evaluations, scVAE demonstrates to be a biologically meaningful tool for identifying cell embeddings and correcting batch effects while preserving inherent biological knowledge. The SC layer offers a more integrated solution by eliminating the need for an explicit gene selection step, improving the efficiency for high-dimensional data processing and streamlining the overall data processing workflow. Furthermore, the learned sparse representation can be utilized to interpret gene informativeness and discover potential marker genes. The BM module removes batch effects by calculating correction vectors. The learned batch-specific variations also empower the batch style transfer feature, broadening the potential applications of the scVAE model. A few ongoing improvements to the implementations are currently in progress to finalize the scVAE model.

## MEANINGFULNESS STATEMENT

This work introduces sparsely connected layers and batchified modules to encode key variations in single-cell RNA-seq data using a self-supervised approach. A series of experiments demonstrate that these building blocks produce biologically meaningful results. The presented `scVAE` model not only outperforms popular methods in the community, but also offers cascading improvements to existing approaches. These advantages streamline the data processing workflow and accelerate the biological discovery process.

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

# A    APPENDIX

## A.1    DATA PREPARATION

Data preparation involves necessary steps before modelling the data with the `scVAE` model. This includes data pre-processing steps to remove low-quality measurements and data augmentation steps to improve model robustness.

### A.1.1    DATA PRE-PROCESSING

To streamline scRNA-seq data processing workflow for downstream biological research, this work adopts a minimal three-step data pre-processing and normalization process following the standard practices in the state-of-the-art methods prior to the neural network model. The steps below are implemented for the integer-valued scRNA-seq count matrix:

1. Genes detected in fewer than $c_{\min}$ cells are removed.

2. Cells containing fewer than $g_{\min}$ detected genes are removed.

3. Logarithm transformation (`log1p`) is applied to the filtered count matrix.

In steps 1–2, hyperparameters $c_{\min} = 3$ and $g_{\min} = 200$ are used. The resultant matrix will be used as the `scVAE` model input with no further data pre-processing required. Specifically, gene selection, such as highly variable genes (HVGs), is not necessary during data pre-processing since the `scVAE` model is designed to operate on the full gene list. The processed count matrix is stored as an `AnnData` object, and the steps above are implemented using the `scanpy` package.

### A.1.2    DATA AUGMENTATION

Depending on the tissue sample, experimental design, and sequencing platform, a single data batch may contain a limited number of cells, making it challenging to learn a robust representation of the data. Furthermore, when multiple batches are presented, the number of cells may vary across batches, leading to insufficient modelling for some batches. To alleviate this issue, several data augmentation strategies are implemented to enrich the original dataset. To avoid potential confusion, this work uses the term *batch* to describe a single piece of sequencing data, while *mini-batch* refers to the number of data points used for training a neural network at a time.

#### SAMPLE AUGMENTATION

Conventional data loaders draw mini-batches sequentially in the training stage, corresponding to the specified mini-batch size. This results in the same mini-batch permutation being repeated across epochs. To improve the training robustness and model performance, instead of drawing the same mini-batch of samples, a random set of samples is drawn with replacement from the entire dataset. This strategy generates a new mini-batch of samples at each step, improving the overall training robustness. However, this strategy weakens the concept of an epoch, as the same training samples may appear multiple times across mini-batches or even within the same mini-batch. For clarity, an epoch in this work refers to the model seeing the same number of samples as the dataset. On average, each sample has an equal probability of being selected at each training step, and the model will be trained on each sample for an equal number of times.

#### SEQUENCING DEPTH AUGMENTATION

The sample augmentation strategy increases overall training randomness by generating different permutations of samples, but the samples remain the same. Therefore, a sequencing depth augmentation strategy is implemented to create more diverse mini-batches. This strategy simulates the same cell being captured with different sequencing depths, reflected as different total molecule counts. For each mini-batch drawn with the sample augmentation strategy, a scalar multiplicative factor is drawn uniformly and independently for each cell from the interval $[0.98, 1.02)$ and is applied to the raw count prior to the logarithm transformation in the data pre-processing steps. This approach creates more diverse training samples while preserving the identity of each cell.

### BATCH AUGMENTATION

When multiple batches are present in the dataset, a batch augmentation strategy is implemented to minimize the bias introduced by varying numbers of cells across batches. This strategy employs the same random sampling with replacement method and draws an equal number of samples from each batch. Furthermore, sample augmentation and sequencing depth augmentation are also applied in addition to the batch augmentation.

The final mini-batch for model training is generated with the combination of the outlined data pre-processing and data augmentation strategies.

### A.1.3 DATASET

Results discussed in this work are based on a TNBC dataset taken from a research investigating stromal cell heterogeneity in human triple-negative breast cancer (TNBC) by Wu et al. (2020). Breast tumour samples were collected from five donors with confirmed TNBC and sequenced with Illumina NextSeq 500. This dataset contains 22,841 genes, 20 cell type groups across 23,917 cells, and spans 5 donors labelled as *p1–5*. Cell type labels were manually annotated using canonical markers. ENA: PRJEB35405, *doi*: 10.15252/embj.2019104063. For single-batch experiments, the model is evaluated on the largest batch within the dataset. Additional experiments are currently underway, including a pancreas dataset by Baron et al. (2016) and an influenza A virus dataset by Madrid et al. (2025).

### DATASET OVERVIEW

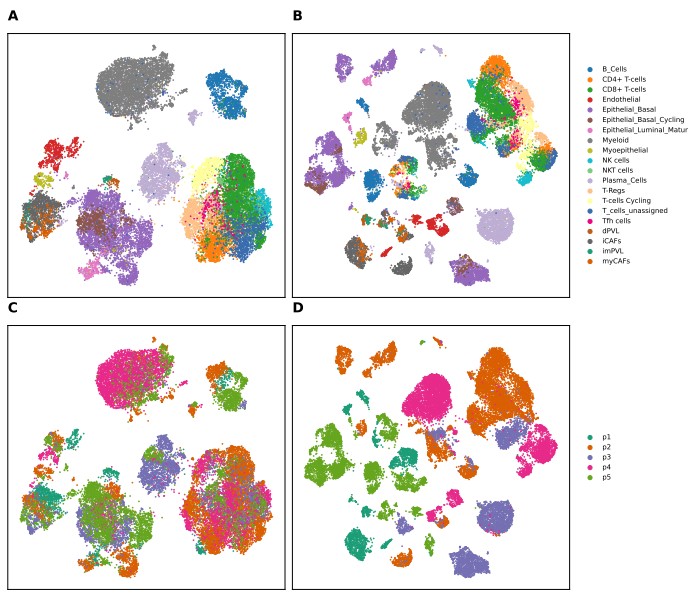

Figure 5: UMAP visualization of multi-batch cell embedding experiment.

**A**: `scVAE` embedding coloured by annotated cell type. **B**: Batch-uncorrected embedding coloured by annotated cell type. **C**: `scVAE` embedding coloured by batch label. **D**: Batch-uncorrected embedding coloured by batch label.

### A.2 IMPLEMENTATION DETAILS OF THE SPARSELY CONNECTED LAYER

The SC layer shares similarities with a classic FC layer and can seamlessly replace a FC layer in existing models. A SC layer is parameterized by two sets of parameters: the weight parameters $\{\mathbf{W}_{\mathrm{SC}}, \mathbf{b}_{\mathrm{SC}}\}$ and the connection parameters $\{\mathbf{M}_{\mathrm{SC}}\}$. For a SC layer with $d_{\mathrm{in}}$ input neurons and $d_{\mathrm{out}}$ output neurons, the corresponding weight parameters are $\mathbf{W}_{\mathrm{SC}} \in \mathbb{R}^{d_{\mathrm{in}} \times d_{\mathrm{out}}}$ and $\mathbf{b}_{\mathrm{SC}} \in \mathbb{R}^{d_{\mathrm{out}}}$, and

the connection parameters are represented by a binary matrix $\mathbf{M}_{\mathrm{SC}} \in \{0, 1\}^{d_{\mathrm{in}} \times d_{\mathrm{out}}}$. All weight and connection parameters of the SC layer are initialized randomly with the same number of connections from each input to the output neurons. The number of initial connections is denoted as $n_{\mathrm{init}} = \mathtt{floor}(\alpha_{\mathrm{init}} \cdot d_{\mathrm{out}})$, where $0 < \alpha_{\mathrm{min}} \le \alpha_{\mathrm{init}} \le \alpha_{\mathrm{max}} < 1$ represents the percentage of connections with maximum $\alpha_{\mathrm{max}}$ and minimum $\alpha_{\mathrm{min}}$.

### A.2.1 CONNECTION ADJUSTMENT SCHEME

Learning SC layer parameters involves two iterative stages. Stage I – weight updates: the connection parameters $\mathbf{M}_{\mathrm{SC}}$ are fixed, and weight parameters $\{\mathbf{W}_{\mathrm{SC}}, \mathbf{b}_{\mathrm{SC}}\}$ are updated. Stage II – connection updates: the weight parameters $\{\mathbf{W}_{\mathrm{SC}}, \mathbf{b}_{\mathrm{SC}}\}$ are fixed, and connection parameters $\mathbf{M}_{\mathrm{SC}}$ are updated. Stage I is accomplished using the standard back-propagation algorithm, which updates weight parameters along with other neural network parameters. Stage II employs several connection adjustment schemes, with details provided below:

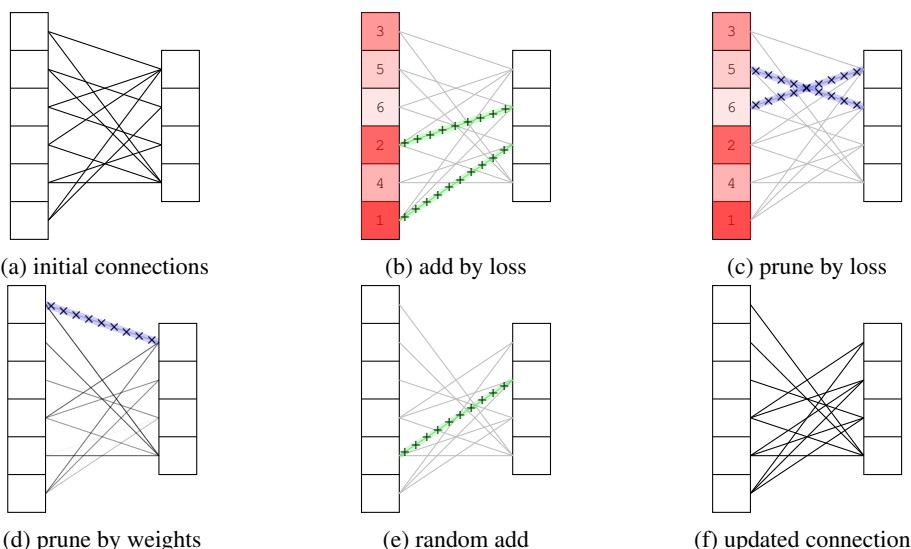

Figure 7: Connection update schemes of the SC layer.

(a) **Initialization**: The connections are initialized with the same number of connections for each input neuron to the output neuron. In this illustrative example, $d_{\mathrm{in}} = 6, d_{\mathrm{out}} = 4, \alpha_{\mathrm{init}} = 0.5, n_{\mathrm{init}} = 2$. This balanced initialization ensures that each input neuron has an equal number of connections at the beginning of the training procedure.

(b) – (c) **Adjustment by loss**: The scVAE model builds on the VAE architecture with a variational bottleneck layer and symmetrical reconstruction layers. The first adjustment scheme evaluates the reconstruction quality of each input gene based on the assumption that, with the presence of a bottleneck layer, if the model poorly reconstructs an input gene, it lacks sufficient information from the input about this gene. Conversely, a good reconstruction indicates that the model has already had enough information. To evaluate and rank the reconstruction quality, the mean-squared error (MSE) is calculated between the raw count value and the mean of the ZINB distribution. For genes with low MSE values, a single connection is removed randomly from existing connections. For genes with high MSE values, a single connection is added randomly to unconnected outputs. Both addition and removal are constrained by the $\alpha_{\mathrm{max}}$ and $\alpha_{\mathrm{min}}$ limits, and no adjustments will be made if the resulting number of connections violates these limits. In each iteration, a single connection will be added for a maximum of $n_{\mathrm{al}}$ genes and a single connection will be removed for a maximum of $n_{\mathrm{pl}}$ genes.

(d) **Prune by weights**: The second adjustment scheme relies on the magnitude of the weight parameter $\mathbf{W}_{\mathrm{SC}}$. A smaller magnitude of the weight potentially reflects less informative and less important connections. Therefore, pruning such connections has limited impact on

the final results. To prune by weights, the absolute value of weights in $\mathbf{W}_{\mathrm{SC}}$ is calculated for all genes. The prune cutoff threshold is determined by the minimum of the $q_{\mathrm{w}}$ quantile and the absolute value cutoff $v_{\mathrm{w}}$. A single connection whose weight is below the cutoff threshold is randomly removed. This adjustment scheme is also constrained by the $\alpha_{\max}$ and $\alpha_{\min}$ limits. A maximum of $n_{\mathrm{pw}}$ connections will be removed in each iteration.

(e) **Random adjustment**: The third adjustment scheme uses randomness to introduce new connections in the SC layer. The randomness serves two purposes: first, it introduces additional randomness in the training procedure, reducing the likelihood of the training being stuck in local minima due to connection initialization. Second, it acts as a garbage collector, ensuring the total number of connections remains the same after an iteration of adjustment. This random adjustment scheme is still constrained by the limits, and if an input gene has saturated in its number of connections, no new connections will be added. The random adjustment iterates through all input genes until the required number of additions is reached. The total number of connections added is determined by the difference between the removed and added connections from the two adjustment schemes described above.

(f) **Iteration**: After adjusting connections under the three schemes outlined above, an iteration of connection update is complete. The updated SC layer has the same number but a different set of connections. These updated connections will be used in the upcoming training epochs for weight updates.

### A.2.2 IMPLEMENTATION

The SC layer is employed as the first encoder layer and the last decoder layer in the `scVAE` model, where the input and output are in the entire gene space, replacing the corresponding conventional FC layers in a vanilla VAE model. While a complete hyperparameter sweep was not conducted, the following hyperparameters are used in the SC layer: $\alpha_{\mathrm{init}} = 0.2, \alpha_{\max} = 0.5, \alpha_{\min} = 0.05, n_{\mathrm{al}} = 0.06 d_{\mathrm{in}}, n_{\mathrm{pl}} = 0.08 d_{\mathrm{in}}, n_{\mathrm{pw}} = 0.02 d_{\mathrm{in}}, q_{\mathrm{w}} = 0.1\%, v_{\mathrm{w}} = 10^{-5}$. The decoder SC layer is operated in the *mirroring* mode, where the binary mask $\mathbf{M}_{\mathrm{SC}}^{\mathrm{dec}} = (\mathbf{M}_{\mathrm{SC}}^{\mathrm{enc}})^{\top}$ is the transpose of the encoder mask after each connection adjustment. Whereas, the decoder SC layer learns its own weight parameters $\mathbf{W}_{\mathrm{SC}}^{\mathrm{dec}}$.

The SC layer is implemented as a subclass of a conventional FC layer with an additional binary mask as described in equation (1). An optional FC layer can be added after the SC layer to further multiplex the output and accommodate the potential difference in reception field for each neuron. While the SC layer shares the same number of weight parameters as a FC layer, the effective number of connections is determined by the $\alpha_{\mathrm{init}}$ parameter and the learned binary mask $\mathbf{M}_{\mathrm{SC}}$. The connection adjustment schemes are implemented in the same order as they are presented in Figure 7.

In a multi-batch experiment, referencing the concept of batch-conserved and batch-variable genes, an individual binary mask $\mathbf{M}_{\mathrm{SC}}^{b}$ is learned for each batch sharing the same weight parameters $\{\mathbf{W}_{\mathrm{SC}}, \mathbf{b}_{\mathrm{SC}}\}$ for both encoder and decoder. Connection updates are performed for each batch individually.

For clarity, this work uses $\bowtie$ to denote a conventional FC layer and $\prec$ to denote the SC layer, where applicable.

### A.3 IMPLEMENTATION DETAILS OF THE BACHIFIED MODULE

### A.3.1 IMPLEMENTATION

Figure 9 illustrates an example of a three-batch experiment denoted by green, blue, and red colours. The sub-modules `M1`, `M2`, and `M3` learn batch-specific variations. When implementing `scVAE`, a pair of BM modules is injected into the encoder (eBM) and decoder (dBM) symmetrically. The eBM module computes the batch correction vectors based on equation (2), such that the output of the eBM module is batch-corrected. Conversely, the dBM module *re-introduces* the batch effect to data whose input is batch-corrected but output is batch-uncorrected. The output of the dBM module will be used to reconstruct the original input data, where the batch effect is present. $\beta_{\mathrm{bm}} = 1$ is used for both the eBM module and the dBM module.

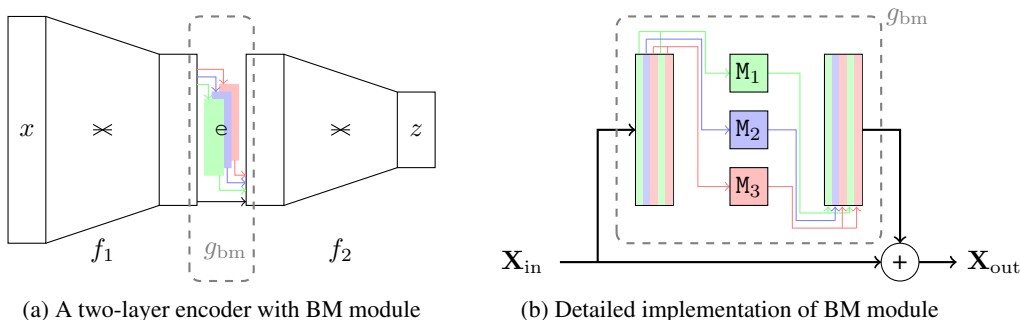

(a) A two-layer encoder with BM module      (b) Detailed implementation of BM module

Figure 9: Detailed implementation of the batchified module.

## A.4 DETAILS OF THE BATCH STYLE TRANSFER EXPERIMENT

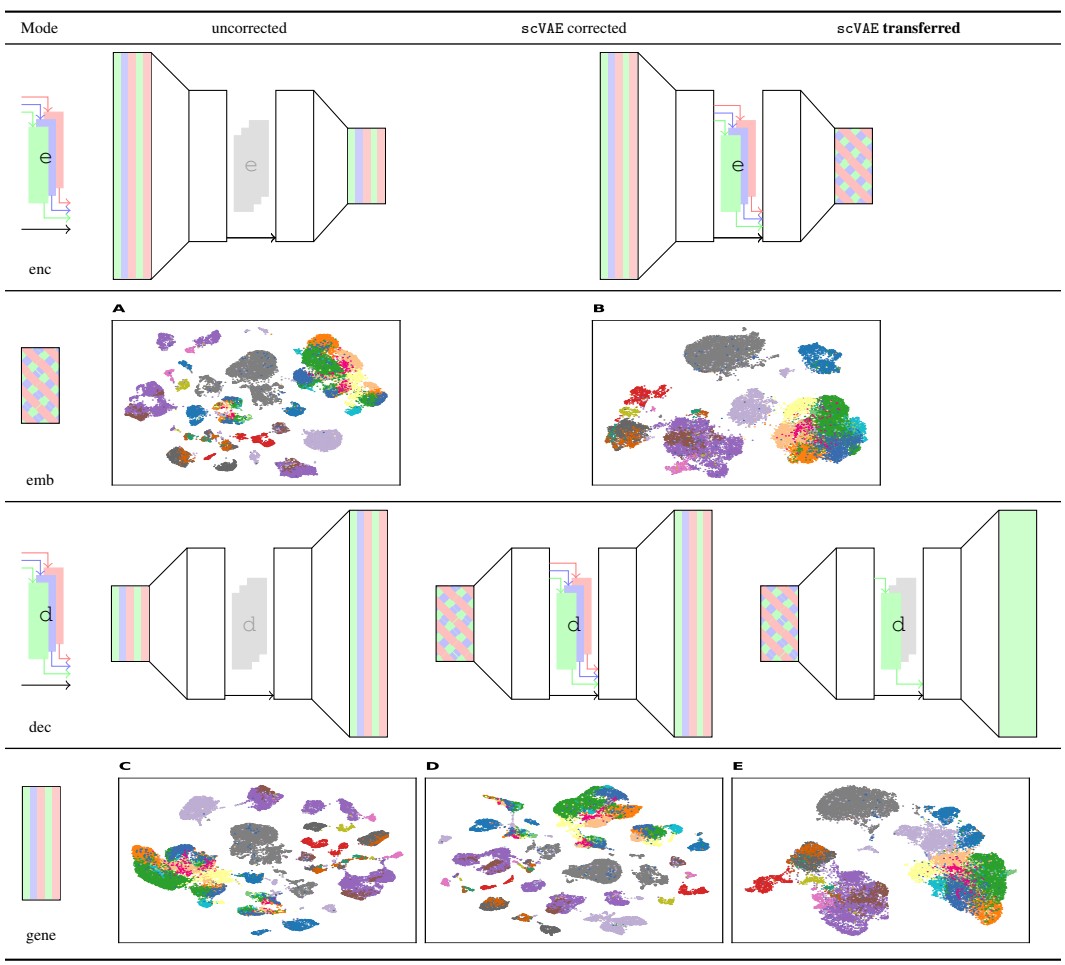

Table 5: `scVAE` **modes and batch style transfer results**.

**A–B**: UMAP visualizations of the embedding space. `scVAE` batch correction and batch transfer share the same encoder model and therefore produce the same embedding. **C–E**: UMAP visualizations of the 50D PCA space of the full gene space.

