# OpenReview forum: "Sparsely Connected Variational Autoencoder for scRNA-seq Data Processing"
_ICLR.cc/2026/Workshop/LMRL — Submitted to ICLR 2026 Workshop LMRL_

### Official Review · Reviewer_i6D4 · 2026-02-23
**Insufficient motivation and evaluation for a sparse VAE for single cells**

**Rating:** 2
**Confidence:** 5

**Review:**

The work proposes a VAE-based framework operating on the full gene space as an alternative to explicit gene selection. It introduces two layer architectures, a sparsely connected layer and a batch-specific layer. It is evaluated on a single dataset against scVI and Seurat using clustering metrics, gene relevance scoring, and embedding visualizations.

This work suffers from significant flaws that include lack of motivation, poor acknowledgement of the current literature, and a weak experimental benchmark.

The need for a dedicated sparse layer architecture to allow VAEs for single cells to handle complete genomes is not clear. Off-the-shelf VAEs have a modest memory footprint during training (a few 100s MB to a few GBs), making these models trainable even on modest hardware. They are also routinely applied in application papers without gene selection.

Related literature that introduced sparse or lightweight generative models is not acknowledged or compared against (see, for instance, Svensson et al., 2020, Lotfollahi et al., 2023, Kunes et al., 2023, Doncevic et al., 2023). The positioning of the work relative to the exhaustive literature on batch effects for single-cell RNA-seq is missing altogether.

Beyond this, the methodology is not described clearly enough to be properly reviewable. For instance, the activation function defining the sparsely connected layer in Equation 1 is not specified. Similarly, the experimental benchmark suffers from notable weaknesses, including, among other things, too few baselines (even for a short paper), an overreliance on visual inspection to appreciate embedding quality, misleading use of significant figures across experiments, and evaluation on a single dataset.

In its current form, this work does not sufficiently motivate or differentiate itself from existing approaches, and the experimental evidence is too limited to support the claims made.

---

### Official Review · Reviewer_CvGa · 2026-02-25

**Rating:** 4
**Confidence:** 3

**Review:**

The authors propose scVAE, a sparsely connected variational autoencoder that integrates multiple scRNA-seq preprocessing tasks into a unified, self-supervised deep learning framework, addressing limitations of traditional multi-stage workflows that require sequential statistical adjustments. By operating on the full gene space without gene selection and modeling batch-specific variation through parameterized correction modules, scVAE reduces processing overhead while improving efficiency, interpretability, and biological consistency across real datasets.

Advantages:
- The architecture includes task-specific modifications tailored to single-cell data: a sparsely connected layer enables modeling of the full gene space with less model parameters, and a dedicated batchified module explicitly addresses batch effects in multi-batch experiments.
- The manuscript provides a detailed and transparent data preparation section, which improves reproducibility and clarifies how preprocessing choices interact with the proposed model.

Disadvantages:
- There is no ablation or direct comparison against a standard VAE baseline without the sparsely connected layer and batchified module, making it difficult to isolate the contribution of each architectural component.
- For multi-batch settings, the evidence supporting the batch-correction mechanism relies primarily on qualitative UMAP visualizations, without quantitative batch-mixing or integration metrics presented in tabular form to substantiate the claims.

---

### Meta-Review · Area_Chair_5XME · 2026-02-25

**Recommendation:** Reject
**Confidence:** 5

**Metareview:**

This paper does not pass the bar for the LMRL workshop

---

### Decision · Program_Chairs · 2026-03-02

**Decision:**

Reject

**Comment:**

Please see the meta-review.